# Magnolol and Luteolin Inhibition of α-Glucosidase Activity: Kinetics and Type of Interaction Detected by In Vitro and In Silico Studies

**DOI:** 10.3390/ph15020205

**Published:** 2022-02-08

**Authors:** Francine Medjiofack Djeujo, Eugenio Ragazzi, Miriana Urettini, Beatrice Sauro, Elena Cichero, Michele Tonelli, Guglielmina Froldi

**Affiliations:** 1Department of Pharmaceutical and Pharmacological Sciences, University of Padova, 35122 Padova, Italy; francine.medjiofackdjeujo@phd.unipd.it (F.M.D.); eugenio.ragazzi@unipd.it (E.R.); miriana1896@gmail.com (M.U.); beatricesauro8@gmail.com (B.S.); 2Department of Pharmacy, University of Genova, 16128 Genova, Italy; tonelli@difar.unige.it

**Keywords:** natural polyphenols, α-glucosidase inhibitors, magnolol, luteolin, enzymatic kinetics, circular dichroism, molecular docking, diabetes mellitus, hyperglycaemia

## Abstract

Magnolol and luteolin are two natural compounds recognized in several medicinal plants widely used in traditional medicine, including type 2 diabetes mellitus. This research aimed to determine the inhibitory activity of magnolol and luteolin on α-glucosidase activity. Their biological profile was studied by multispectroscopic methods along with inhibitory kinetic analysis and computational experiments. Magnolol and luteolin decreased the enzymatic activity in a concentration-dependent manner. With 0.075 µM α-glucosidase, the IC_50_ values were similar for both compounds (~ 32 µM) and significantly lower than for acarbose (815 μM). Magnolol showed a mixed-type antagonism, while luteolin showed a non-competitive inhibition mechanism. Thermodynamic parameters suggested that the binding of magnolol was predominantly sustained by hydrophobic interactions, while luteolin mainly exploited van der Waals contacts and hydrogen bonds. Synchronous fluorescence revealed that magnolol interacted with the target, influencing the microenvironment around tyrosine residues, and circular dichroism explained a rearrangement of the secondary structure of α-glucosidase from the initial α-helix to the final conformation enriched with β-sheet and random coil. Docking studies provided support for the experimental results. Altogether, the data propose magnolol, for the first time, as a potential α-glucosidase inhibitor and add further evidence to the inhibitory role of luteolin.

## 1. Introduction

The pathological importance of diabetes mellitus (DM) is linked to its widespread diffusion throughout the world, its chronic course, and its short- and long-term complications in humans. Despite therapeutic progress, DM remains a key contributor to chronic renal failure and blindness in adults and can lead to amputation of the lower limbs; moreover, it represents one of the main risk factors for various cardiovascular diseases [1]. α-Glucosidase is an enzyme that belongs to the hydrolase class and catalyzes the breakdown of complex carbohydrates into monosaccharides; therefore, it plays an important role in glycemic control. One of the current therapeutic approaches to reduce postprandial hyperglycemia is based on the use of α-glucosidase inhibitors, such as acarbose, miglitol, and voglibose [2]. However, these drugs have shown various side effects; therefore, great attention is needed to find new α-glucosidase inhibitors endowed with improved pharmacological profiles. In this regard, several studies suggested that plant-derived products offer a great perspective for the prevention of DM and its complications without relevant side effects [3]. Among various chemical classes of natural compounds, polyphenols have been shown to inhibit α-glucosidase enzyme activity and glycation processes [4,5]. Indeed, in vitro and in silico structure-activity relationship investigations have demonstrated that flavonoids possess promising α-glucosidase inhibitory potential [6], suggesting that further study on this topic could be of interest.

In this study, magnolol and luteolin were evaluated and compared with acarbose (Figure 1) on the selected target α-glucosidase, involved in the control of hyperglycemia and related induced damage. Magnolol is a polyphenol belonging to the class of lignans, detected mainly in *Magnolia officinalis* [7]. Indeed, several plant-derived preparations obtained from various species of *Magnolia* are widely used in traditional Chinese medicine to obtain antiproliferative, anti-inflammatory, antioxidant, and antidepressant effects [8]. Furthermore, these preparations are also used for their beneficial effects on the gastrointestinal, respiratory, and cardiovascular systems [7]. Interestingly, magnolol was shown to preserve and improve the function of RIN-m5F β-pancreatic cells by increasing the gene expression of PDX1 (pancreatic and duodenal home box (1) and glutathione peroxidase) [9]. Although some authors described in cell cultures and in vivo models the hypoglycemic activity of magnolol [9,10,11,12], its effect on α-glucosidase has not been studied yet. Luteolin is a well-known flavonoid, detected in several medicinal plants. It is recognized for its antioxidant and anti-inflammatory properties, its protective role in cardiovascular diseases, and its preventive function in cancer development [13,14,15,16]. Furthermore, the ability of luteolin to target α-glucosidase has also been described [17]. However, for a more in-depth study of the inhibitory action of magnolol, a careful comparison with those of luteolin and acarbose is provided. These investigations were accompanied by homology modeling and docking studies that allowed elucidating at, the molecular level, how compounds are capable of damping α-glucosidase activity. These results are expected to be useful for the future development of new α-glucosidase inhibitors helpful for hyperglycemic control.

## 2. Results and Discussion

### 2.1. Yeast α-Glucosidase Inhibition

Magnolol and luteolin were evaluated for their inhibitory activity on α-glucosidase, including acarbose as positive control. Magnolol and luteolin were tested from 5 to 100 μM on 0.075 μM α-glucosidase with 2 mM pNPG as substrate. Both compounds exhibited a marked concentration-dependent inhibition (Figure 2). The IC_50_ values of magnolol and luteolin were 32.6 μM and 32.3 μM, respectively, 25 times lower than those of acarbose (IC_50_ = 815.4 μM, Table 1). Thus, in this experimental condition, magnolol and luteolin showed equivalent inhibitory potency. Other authors performed similar evaluations with luteolin also using acarbose as positive control; examples of reported IC_50_ values for luteolin are 13.07 μM [18] and 46 μM [6], while for acarbose they are 228.16 μM [18] and 606 μM [6], showing that luteolin was, respectively, 13 and 17 times more potent than acarbose.

To explore the type of inhibition, magnolol and luteolin were studied at different enzyme and substrate concentrations, and their activities were compared to those of acarbose. Figure 3 reports how the enzymatic activity changed at different α-glucosidase concentrations, maintaining a constant amount of chromogenic substrate for the enzyme (2 mM pNPG). As expected, the absorbance curves shifted to the left with higher enzyme concentrations (Figure 3A), and consequently the relative rate of reaction increased (Figure 3B).

As a next step, the study was extended to magnolol, luteolin, and acarbose with different enzyme concentrations (0.025, 0.05, 0.075, and 0.125 μM). Figure 4 shows the kinetic curves obtained with 0.075 μM α-glucosidase and 2 mM pNPG with different concentrations of each inhibitor. It can be observed that as the concentration of the inhibitor increases, the speed of the reaction decreases, causing a longer time to yield the final product. Table 1 shows the IC_50_ values of each inhibitor against different concentrations of α-glucosidase. 

### 2.2. α-Glucosidase Inhibition: A Reversible Interaction

To better understand the type of enzyme-inhibitor interaction, the graph “v vs. [α-glucosidase]” was obtained, using different concentrations of magnolol, luteolin, and acarbose (Figure 5). These were responsible for a reversible inhibition of the enzyme because all straight lines intercept the origin with a decreasing slope at increasing concentrations of each inhibitor. This pattern is in agreement with previous findings for luteolin and acarbose [17,19], while this is the first evidence for magnolol of its reversible interaction with α-glucosidase.

### 2.3. α-Glucosidase Inhibition: Inhibitory Kinetic Analysis

The type of inhibition of magnolol, luteolin, and acarbose was estimated using Michaelis–Menten and Lineweaver–Burk plots (Figure 6). 

The inhibition constants (K_i_) of magnolol, luteolin, and acarbose were obtained by secondary plots of “slope *versus* [I]”, Table 2. Magnolol and luteolin were at least 5 and 10 times more potent than acarbose. Furthermore, the secondary slope plots of magnolol, luteolin, and acarbose were linear (Figure 6C,F,I), suggesting that the inhibitors have a single site of interaction or a single class of inhibition sites on α-glucosidase [20]. For magnolol, the inhibition constant of the enzyme (K_i_) and the inhibition constant of the enzyme-substrate complex (K_i′_) were, respectively, 67.0 µM and 160.4 µM, showing a higher potency than acarbose (K_i_ = 356.3 µM). However, magnolol showed a weaker inhibitory strength compared to luteolin, whose K_i_ and K_i’_ values were 34.2 µM and 35.4 µM, respectively (Table 2). Similar results for luteolin were previously reported showing K_i_ = 29 µM and K_i’_ = 38 µM [4], while other authors described for luteolin a K_i_ of 1.40 ± 0.02 *×* 10^−4^ mol/L [17]. Dissimilar K_i_ values could be due to different experimental protocols [21]. The catalytic efficiency of α-glucosidase expressed as K_cat_/K_m_ was also calculated (Table 2). It can be observed that K_cat_/K_m_ decreased mainly in the presence of magnolol and luteolin, suggesting a decrease in the catalytic efficiency of α-glucosidase. The K_i_ and K_i’_ values were almost equal for luteolin, suggesting a similar affinity to bind the free enzyme or the enzyme-substrate complex (non-competitive inhibition) [21]; conversely, magnolol preferentially bound the free enzyme, since K_i_ was lower than K_i‘_. Otherwise, acarbose acts exclusively on the free enzyme, while competing with the substrate to access the binding site of the α-glucosidase. Furthermore, the number of times the enzyme restarts its cycle per unit of time (K_cat_) is nearly constant with different concentrations of acarbose, while it decreases with increasing concentrations of luteolin and magnolol. Additionally, K_cat_ values in the presence of luteolin had a faster decline compared to those of magnolol. In general, the maximal velocity (V_max_) of the reaction in the presence of magnolol decreased slowly and progressively (Appendix A), while that of luteolin decreased quickly (Appendix A), whereas that of acarbose remained almost constant (Appendix A). The K_m_ and V_max_ of α-glucosidase at different concentrations of magnolol, luteolin, and acarbose are reported in the Appendix A. From the analysis of the Lineweaver–Burk graphs, it can be observed that for magnolol, all data lines intersected the x-axis in the second quadrant, indicating a mixed-type inhibition. In fact, both V_max_ and K_m_ changed simultaneously at different concentrations of magnolol, defining a typical mixed-type mechanism (Appendix A). 

The trend of the kinetic parameters reveals three different types of inhibition mechanisms, where acarbose behaved as a competitive inhibitor, magnolol as a mixed-type inhibitor, and luteolin as a non-competitive inhibitor. Actually, the type of interaction of luteolin highlighted in the current investigation is consistent with previously reported data [6,17,21]. Regarding acarbose, unsaturated cyclitol has been described to be responsible for its competitive inhibition because it interacts with the same binding site of the substrate [22].

### 2.4. Inactivation Kinetics, Time Course and Thermodynamics

Each inhibitor rapidly reduced enzyme activity at the lowest concentrations tested (10–25 µM magnolol, 5–25 µM luteolin, and 200–1600 µM acarbose) during the first 300 s, then the catalytic activity was almost constant until 1500 s (Figure 7). The semilogarithmic graphs of Figure 7 indicate a monophasic process for magnolol and luteolin, as well as for acarbose, in the concentration range studied. For luteolin, this result is similar to that already reported in the literature [17]. Thus, the inactivation process in the presence of magnolol and luteolin followed first-order kinetics. Furthermore, the transition free energy and the inactivation kinetic constant (k) were obtained (Table 3). It can be observed that K values increased along with inhibitor concentrations, while the transition free energy decreased in a concentration-dependent manner for both magnolol and luteolin, similarly to that for acarbose. These data indicate that the inhibitors quickly and spontaneously bind to the enzyme, deactivating its catalytic function. Furthermore, the kinetics of the interaction show that the process reaches an equilibrium state in a very short time, suggesting specific binding sites for inhibitors on α-glucosidase [23].

### 2.5. Interaction Characteristics between Inhibitors and α-Glucosidase

#### 2.5.1. α-Glucosidase Fluorescence Quenching by Magnolol, Luteolin, and Acarbose

Figure 8 shows the high fluorescence intensity of α-glucosidase at 337 nm when excited at 280 nm, at 298 K (pH = 6.8). The Stern–Volmer constant (K_sv_) and bimolecular quenching constant (K_q_) were determined using the graphs “F_o_/F vs. [Q]” (Figure 8D–F, and Table 4). The plots obtained at 298, 304, and 310 K show a linear fit, suggesting a single quenching mechanism. The K_sv_ and K_q_ values of magnolol are positively correlated with the temperature, while those of luteolin and acarbose are inversely associated with it (Table 4). Furthermore, the K_q_ values of each compound were three orders of magnitude higher than the maximum scatter collision quenching constant (2 × 10^10^ M**^−^**^1^sec**^−^**^1^) [24].

Based on the K_sv_ and K_q_ values, it is possible to assume that the quenching of α-glucosidase by magnolol is a static-dynamic quenching process [25,26]. On the other hand, the static quenching mechanism is the predominant way of interaction between luteolin and α-glucosidase, in agreement with previously reported data [17]. Furthermore, the K_sv_ values of magnolol and luteolin were generally higher than those of acarbose, suggesting that the fluorescence quenching activity induced by the two polyphenols on α-glucosidase is more powerful than that of acarbose [27]. 

#### 2.5.2. Thermodynamic Parameters and Nature of Binding Forces

The binding constant (K_a_) and the number of binding sites per protein (n) for the α-glucosidase-inhibitor system with magnolol, luteolin, and acarbose were estimated from the intercept and slope of the double logarithmic regression curve (Appendix A). For the magnolol-α-glucosidase system, the binding affinity increased with increasing temperature, while for the luteolin and acarbose-protein systems the binding affinity decreased at higher temperatures (Table 5). This trend agrees with Stern–Volmer information and could imply changes in the tertiary structure of protein folding [28]. The number of binding sites in the presence of magnolol was substantially 0.6, while it was approximately 1 in the presence of luteolin and acarbose. Furthermore, the number of binding sites for magnolol and luteolin increases with temperature. The data obtained with luteolin converge with those previously described, suggesting the presence of a single binding site [17].

Thermodynamic parameters provide information on the characteristics of the bonds of a reaction. Therefore, the binding forces between α-glucosidase and the inhibitors were studied through the evaluation of the variation of Gibbs free energy (ΔG°) which depends on changes in enthalpy (ΔH°) and entropy (ΔS°), according to the following equation: ΔG° = ΔH° − TΔS°. Negative free energy variation ΔG° indicates that the process is spontaneous, while positive ΔG° means that the process is nonspontaneous. Therefore, the negative ΔG° values of magnolol and luteolin with α-glucosidase indicate that the interaction reactions were spontaneous (Table 5). Furthermore, the positive value of ∆S° (121.83 J mol**^−^**^1^ K**^−^**^1^) of magnolol suggests that hydrophobic interactions play a predominant role in its entropy-driven reaction [29]. The negative ΔS° and ΔH° of luteolin and acarbose suggest that van der Waals and the H-bond drove their interaction with the enzyme [29]. The ΔG° of acarbose was positive while the ΔH° and ∆S° were negative, indicating that the interaction between acarbose and α-glucosidase is also spontaneous at lower temperatures [30]. The binding constant (K_a_) decreased in the order luteolin > magnolol ≥ acarbose at the detected temperatures, indicating that luteolin had the strongest binding affinity to the enzyme. Furthermore, the K_a_ values of magnolol showed an increasing trend with increasing temperature, indicating that the magnolol-enzyme complexes were more stable at higher temperatures, unlike luteolin and acarbose (Table 5).

#### 2.5.3. Energy Transfer between Inhibitor and α-Glucosidase

Förster resonance energy transfer is a method based on the calculation of energy transfer between an excited-state donor fluorophore and a ground-state acceptor. The quantum yield (φ) of α-glucosidase was used to determine the distance between the donor and the acceptor [30]. The overlap between the fluorescence emission spectrum of α-glucosidase and the UV absorbance spectra of magnolol is reported in Appendix A. The calculated distance between magnolol and α-glucosidase was 2.06 nm. Previously, a value of 4.56 nm was reported for luteolin [17]. The two inhibitor distances are less than 7 nm, which implies that the transfer of energy from α-glucosidase to magnolol, as well as luteolin, occurred with high probability. Furthermore, the data indicate that the energy transfer of the enzyme with inhibitors is a non-radiative transfer process [31,32,33].

### 2.6. Conformational Change of α-Glucosidase

#### 2.6.1. Synchronous Fluorescence Spectra

For a deeper investigation of the interaction between inhibitors and α-glucosidase, synchronous fluorescence spectroscopy was applied [34,35]. Figure 9 shows the synchronous fluorescence spectra of the residues of tyrosine and tryptophan with magnolol, luteolin, and acarbose. The red shift from 286 to 290 nm for magnolol and the blue shift from 286 to 283 nm for luteolin and acarbose (Figure 9A–C) were observed when ∆λ = 15 nm, while there was no modification at ∆λ = 60 nm (Figure 9D–F), suggesting that the enzyme-ligand interaction did not significantly influence the microenvironment of tryptophan. Therefore, the results show a change in the microenvironment of tyrosine, indicating that the inhibitors affected the enzyme structure, leading to exposure to the solvent and subsequent displacement of the tyrosine residues to more hydrophilic residues.

#### 2.6.2. Circular Dichroism (CD) Measurements

Circular dichroism spectroscopy was used to characterize the secondary structure of α-glucosidase and to estimate its changes during interaction with inhibitors [17,36]. The enzyme CD spectra reported two negative minimum values at 209 and 222 nm, matching with the α-helix structure (Figure 10). In the presence of inhibitors, negative humped peaks at 209 and 222 nm decreased compared to α-glucosidase alone. Increasing the molar ratio of magnolol α-helix contents decreased, while β-sheet, β-turn, and random coils increased (Figure 10D–F). Interestingly, magnolol, as well as luteolin, reduced the helicity of α-glucosidase-producing reorganization and conformational changes in the enzyme structure. According to the literature, luteolin also markedly reduced α-helix content while β-sheet and β-turn contents increased [17]. Acarbose, instead, only slightly affected the secondary structure of the enzyme (Figure 10C,F). No published dichroism studies for acarbose have been found.

Synchronous fluorescence data suggested that magnolol and luteolin affect the α-glucosidase structure, conditioning the microenvironment of tyrosine residues. This effect may be related to the change in the structure of α-helix found in the CD study, since tyrosine, with its large side group, is able to destabilize α-helices. Other authors applied CD analysis on α-glucosidase, e.g., with tannins, showing that the interaction of ligands with the enzyme leads to a loss of the secondary α-helix structure, also decreasing biological activity [37]. Together, synchronous fluorescence and CD data provide evidence for the alteration of the enzyme α-helix structure that determines the inhibition of enzymes produced by the investigated compounds.

### 2.7. α-Glucosidase Inhibition: Theoretical Homology Modeling 

Structure-based studies have been performed on the putative binding mode experienced by acarbose, as well as by several series of α-glucosidase inhibitors, based on homology modeling techniques [38,39,40]. Most of them relied on the X-ray crystallographic data of *Saccharomyces cerevisiae* 1,6-glucosidase as a protein template, including maltose (pdb code = 3A4A) [41] and isomaltose (pdb code = 3AXH) [42] as enzyme substrates. Recent efforts have been made to navigate the inhibitory capacity of flavonoids as α-glucosidase targeting compounds [43]. The authors reported that the introduction of the 4**′**-hydroxyphenyl ring as a substituent linked to position 2 of the main chromone core guarantees a higher inhibitory potency value (1, IC_50_ = 8.97 µM, Figure 11) than the 3**′**,4**′**-dihydroxyphenyl analogue (2, IC_50_ = 77.42 µM, Figure 11) [19]. In particular, further substitution at position 3 of the main core with sugar-containing O-galloyl motifs increased the inhibitory effect on the enzyme (3-5, IC_50_ = 0.97-27.84 µM, Figure 11), which proved to be more potent than the previously cited 1,2. Inhibition kinetic assays illuminate the non-competitive behavior of 1–4, with compound 5 being only a competitive inhibitor. Interestingly, the presence of the glucopyranose unit (which includes the galloyl group at position 6 of the sugar ring) instead of the galactose moiety (4) made analogue 5 a competitive inhibitor of α-glucosidase [19]. On the contrary, analogue 3 was a non-competitive derivative, functionalized by the galloyl moiety at position 3 of the galactose unit (Figure 11). Molecular docking studies performed by the same authors in tandem with inhibition kinetic assays also revealed the key contacts, supporting the common inhibitor behavior shared by acarbose and 5, thanks to H-bonds with D214, R312, D349, and R439 targeting the orthosteric binding site (OBS). Non-competitive analogues 1–4 target four different allosteric binding sites (ABSs) within the enzyme. Among them, two protein cavities explored by molecular docking calculations led to the lower and then most relevant scoring function values for derivatives 1-4. The two putative ABSs include: (i) D282, A284, T287, F333, and D338 and (ii) E10, W14, K12, K15, H258, G268, E270, I271, and S295. In the present study, protein modeling was carried out with a ligand-based homology (LBH) modeling protocol, referring to the experimental data of 1,6-glucosidase (pdb code = 3AXH) [42], as a template for the in-house α-glucosidase model. This choice allowed us to take into account the structure of the co-crystallized ligand isomaltose during the protein homology modeling calculation [44].

The theoretical model of the target protein, yeast α-glucosidase, was built from alignment of the related FASTA sequence (P38158) with the X-ray coordinates of 1,6-glucosidase chosen as a protein template (pdb code = 3AXH) in the presence of isomaltase [42] (see material and methods section for details). The reliability of the alignment derived was verified by the high value of the pairwise percentage residue identity evaluated between the two enzymes (PPRI = 72%, Appendix A). Consequently, a consistent number of residues were conserved between the two proteins, as reported by superimposition of the modeled protein with respect to the template root mean square deviation value (RMSD), calculated from carbon atom alignment = 0.229 Å (Appendix A). The backbone conformation of the final yeast α-glucosidase model was inspected by the Ramachandran plot, showing the absence of outliers (Appendix A). Following this procedure, while only the enzyme OBS was determined based on the positioning featured by isomaltose, all putative regions prone to be involved in inhibitor contacts, including the most probable ABSs, were identified by the MOE site finder module. This kind of approach was validated by the following molecular docking calculations of α-glucosidase inhibitors 1–5, including sugar and non-sugar derivatives, obtaining comparable results in terms of key contacts and relevant amino acids [19]. 

#### 2.7.1. α-Glucosidase Inhibition: Binding Site Analysis and Molecular Docking Studies

The search of the most probable binding domains within the in-house modeled protein has been managed by MOE Site Finder, suggesting the three top ranked cavities, such as OBS, ABS1, and ABS2. All amino acids included within these three binding cavities are listed in Appendix A. Based on the previously cited experimental data, the docking scoring functions of OBS, ABS1, and ABS2 were considered for the two epimer derivatives 5 and 4, ABS2 being the most probable binding site for this chemotype. According to the calculations, the sugar-containing galloyl derivatives 4 (non-competitive inhibitor) and 5 (competitive inhibitor) interact with the ABS2 and OBS cavities, respectively (Appendix A). In detail, compound 5 was stabilized within the OBS through H-bonds involving: (i) two hydroxyl groups on the phenyl ring in position 2 of chromone and F157, D408; (ii) one of the two hydroxyl groups of the chromone core and D349; and (iii) the sugar-motif and E304 residue. While the chromone ring was placed in stacking with F300, the sugar motif was properly projected toward a mild polar area of the enzyme, including P309, T307, S308, and R312 (Appendix A). Furthermore, the galloyl substituent was surrounded by a deep cavity delimited by F157, L218, P240, N241, and H245, detecting van der Waals interactions. The related analogue 4 was predicted to target ABS2 by featuring several H-bonds between the sugar motif and S329, while the chromone ring was H-bonded to V297, S299, and T342 (Appendix A). Furthermore, one of the two hydroxyl groups in the phenyl ring in position 2 of the chromone core showed an H-bond with Y286 and N283, while the galloyl substituent was projected onto the polar residues H251, H279, and S281, which is H-bound to S281 and N283.

#### 2.7.2. Docking Studies of Non-Sugar-Containing α-Glucosidase Inhibitors: Kaempferol and Quercetin 

The reliability of the theoretical protein model was evaluated for non-sugar containing α-glucosidase inhibitors kaempferol (1) and quercetin (2). The removal of any (O-galloyl)-sugar-containing substituent in the chromone core maintained the main pharmacophore features to achieve inhibitor ability. Flavonoids 1 (IC_50_ = 8.97 µM) and 2 (IC_50_ = 77.42 µM) experienced modest potency as non-competitive inhibitors, being, in any case, one more effective than acarbose (IC_50_ = 50.58 µM) [19]. In particular, based on current molecular docking calculations, 1 and 2 were able to occupy ABS1 and ABS2, the first being preferred in terms of calculated scoring functions (Appendix A). This kind of positioning allowed inhibitor 1 to detect three H-bonds with the enzyme in ABS1, thanks to the hydroxyl group at positions 3, 5, and 7 of the chromone and V294, T287, and E10, respectively (Appendix A). Furthermore, the terminal phenyl ring was involved in contact with H258 and π−π stacking with Y292. Therefore, the chromone core was efficiently stabilized within the enzyme cavity through van der Waals contacts and polar interactions with W14, K15, and W340. On the contrary, the presence of another H-bonding group on the terminal phenyl ring, such as a second hydroxyl moiety of 2, reversed the coupling mode of the inhibitor, guiding 2 in the proximity of E10, K15, and S339 at the expense of the aforementioned contacts featured by the chromone core (Appendix A), especially those involving E10 and V294. Furthermore, only the catechol portion showed H-bonds with T287 and S339. Effectively, 1 (IC_50_ = 8.97 µM) proved to be more potent than 2 (IC_50_ = 77.42 µM). The results of molecular docking obtained in the ABS2 crevice supported the key role played by the chromone ring that anchors the inhibitor at the enzyme binding site (Appendix A). The bicyclic core was H-bonded to L285, Y286, and N283, while the terminal phenyl ring was projected towards H251 and Y286, detecting polar contacts, and π−π stacking. The presence of the catechol moiety impaired the potency of inhibitor 2, H-binding the catechol group to T287: this type of positioning limited the number of polar contacts of the chromone core, which was able to maintain only one H-bond with N283 (Appendix A). This piece of information focused on inhibitors that did not contain sugar and allowed the deepening of molecular docking calculations of magnolol and luteolin.

#### 2.7.3. Docking Studies of Non-Sugar Containing α-Glucosidase Inhibitors: Magnolol and Luteolin

Table 1 shows the IC_50_ values of magnolol and luteolin obtained with different enzyme concentrations; with 0.05 µM α-glucosidase, the IC_50_ values were 28.5 µM and 21.6 µM, respectively. In fact, the α-glucosidase inhibitory activity was lower compared to the testing compounds 1 and 2, also studied with 0.5 µM of α-glucosidase [19]. This outcome is supported by the results of molecular coupling, according to the following main information: (i) magnolol and luteolin did not bind ABS1 due to the unfavorable values of the scoring function, and (ii) the scoring functions accompanying the coupling poses of magnolol and luteolin in ABS2 agree with a limited number of contacts that turn on enzyme inhibition. As shown in Figure 12A, the two inhibitors maintained the key contact previously cited with N283, as experienced by compounds 1, 2, and 4, thanks to one of the phenolic hydroxyl groups. Additionally, magnolol also showed a further H-bond with H279 because the aromatic rings and the allyl groups highly stabilized in the enzyme cavity by stacking and van der Waals contacts with H251, Y286, and F333 (Figure 12B). The presence of a catechol group in the chemical structure of luteolin moved the inhibitor in proximity to N283, Y286, and T287, gaining polar contact with the protein (Figure 12C). This kind of positioning allowed the compound to detect two H bonds with N283 and T287, as well as π−π stacking with Y286.

## 3. Materials and Methods

### 3.1. Reagents

Acarbose, α-glucosidase (EC 3.2.1.20, *Saccharomyces cerevisiae* type I, 10 U/mg protein), dimethylsulfoxide (DMSO), luteolin (3’,4’,5,7-tetrahydroxyflavone), methanol, *p*-nitrophenyl-α-D-glucopyranoside (pNPG), buffered phosphate saline (PBS) and magnolol (5,5′-diallyl-biphenyl-2,2′-diol) were purchased from Merck KGaA, Darmstadt, Germany. The purity of the reference standards was ≥97%, while other chemicals were of at least analytical grade.

### 3.2. Yeast α-Glucosidase Inhibitory Assay 

α-Glucosidase inhibitory activity was determined according to a previous method [45]. Briefly, 0.1 mM PBS (pH 6.8) was used to dissolve α-glucosidase and pNPG. In detail, 80 μL of PBS or 80 μL of test sample were added to 96-well plates. Successively, 20 μL of enzyme (0.025–0.125 µM) were added to each well. Magnolol (10–100 μM) and luteolin (5–50 μM) were prepared in DMSO (<1% *v*/*v*). Acarbose (200–4000 μM) was used as a positive control. Each sample was incubated with α-glucosidase for 10 min at 37 °C. The reaction started by adding 200 μL of pNPG (2 mM). The absorbance values were detected at 405 nm for 60 min, using a PerkinElmer Victor Nivo microplate spectrophotometer (Waltham, MA, USA). The α-glucosidase activity in the absence of inhibitors was defined as 100%. The half-maximal inhibitory concentration (IC_50_) was estimated by plot of relative enzymatic activity vs. inhibitor concentration.

### 3.3. Kinetic Analysis of Yeast α-Glucosidase Inhibition

The type of enzyme inhibition exerted by inhibitors was evaluated from kinetic studies using different substrate concentrations (0.25–2.5 mM pNPG) applying Michaelis–Menten and Lineweaver–Burk plots. Inhibition constants (K_i_) were determined by plots of the Y-intercept of the Lineweaver–Burk plot vs. inhibitor concentration.

To evaluate the kinetics of inactivation and the rate constant, the time course curves of α-glucosidase in the presence of magnolol (10–100 µM), luteolin (5–50 µM), and acarbose (200–4000 µM) were studied. The enzyme and substrate concentrations were 0.075 µM and 2 mM, respectively. 

### 3.4. Interaction Characteristics between Inhibitors and Yeast α-Glucosidase 

#### 3.4.1. Fluorescence Quenching Analysis

The interaction of inhibitors with α-glucosidase was studied using the fluorescence quenching method. Therefore, the fluorescence of α-glucosidase alone and in the presence of magnolol, luteolin, and acarbose was studied at different concentrations (Jasco FP-6500 spectrofluorometer, Japan). Measurements were made in the emission range of 300–450 nm, with an excitation of 280 nm, after 10 min of stabilization. The fluorescent spectra of α-glucosidase (0.35 µM) and each inhibitor (0–1.0 µM) were carried out at three different temperatures (298, 304, and 310 K) and the bandwidths were set at 5 nm for both emission and excitation slits. For each sample, three fluorescence spectra were acquired, and the blank was subtracted. The compound quenching mechanism was assessed using the Stern–Volmer equation: F_0_ / F = 1 + K_sv_[Q] = 1 + K_q_**^.^**τo[Q], where F_0_ and F are the fluorescence measure in absence and in presence of different concentrations of quencher, respectively; K_sv_ is the Stern–Volmer quenching constant; K_q_ is the coefficient of quencher rate coefficient; τ_0_ is the life time of the excited state; and [Q] is the concentration of the quencher [46]. 

Further elaboration of the equilibrium within the free and bound molecules can be achieved through the analysis of the fluorescence emission data utilizing the following equation: log (F_0_ − F) / F = log K_a_ + n log [Q], which obtains the binding constant (K_a_) and the number of binding sites per protein (n) for the enzyme-inhibitor system [47]. 

#### 3.4.2. Thermodynamic Parameters and Nature of Binding Forces

Thermodynamic spontaneity is an important factor in the study of protein-ligand interaction. A system tends not only toward a lower energy state (enthalpy, ΔH), but also according to a state of disorder (entropy, ΔS). Thermodynamic parameters were obtained according to the Van ’t Hoff equation and the plot of “log K_a_ vs. 1/T”. The trade-off between enthalpy (ΔH°) and entropy (ΔS°) is used to obtain the standard free energy variation (ΔG°) [48].

#### 3.4.3. Non-Radiation Energy Transfer

The distance between magnolol and α-glucosidase was calculated according to the theory of Förster resonance energy transfer [29,33]. To calculate the distance (r) between the donor and the acceptor, the emission wavelength of α-glucosidase was overlapped with the absorption band of the inhibitor. The fluorescence spectrum and the UV absorption spectrum of the inhibitor were collected in the wavelength range of 300–500 nm, at room temperature, pH 6.8, and at a concentration of 0.35 µM.

### 3.5. Conformational Changes of Yeast α-Glucosidase during Magnolol-Mediated Inhibition

#### 3.5.1. Synchronous Fluorescence Spectra

The application of this method allows the identification of conformational changes in proteins according to the spectroscopic behavior of the tyrosine and tryptophan residues [30]. Synchronous fluorescence spectra were collected in the emission range of 260–320 nm [49]. The difference between excitation and emission wavelength (Δλ) was established at 15 nm (tyrosine residues) or at 60 nm (tryptophan residues) [34]. 

#### 3.5.2. Circular Dichroism Measurements

To investigate changes in the secondary structure of α-glucosidase in the presence of inhibitors, circular dichroism (CD) spectra were collected using the Jasco J-810 circular dichroism spectropolarimeter (Tokyo, Japan), at wavelengths between 200 and 250 nm in a nitrogen environment (1 atm). All solutions were prepared in PBS (pH 6.8). The quartz cuvette used had a path length of 1 cm. The concentration of α-glucosidase was 1 µM, and the molar ratios of the inhibitors were 0:1, 1:1, and 2:1. Blank and PBS signals were removed to produce an accurate background signal. Quantification of the different components of the secondary structure (α-helix, β-sheet, β-turn and random coil) of α-glucosidase was established using the online SELCON3 program [50].

### 3.6. Homology Modeling and Molecular Docking Studies

Compounds were manually built using the MOE Builder program and then parameterized (AM1 partial charge as calculation method) and energy was minimized using the Energy Minimize Program setting the MMFF94x force field and RMS (root mean square) equal to 0.0001 kcal/mol/Å2 of the MOE calculation module, to produce a single low energy conformation for each ligand [51]. Docking calculations within yeast α-glucosidase have been performed based on the modeled biological target. In particular, the theoretical model of *Saccharomyces cerevisiae* α-glucosidase was built based on the FASTA sequence (P38158) downloaded from the SWISS-PROT database [52], while the three-dimensional structure of the template 1,6-glucosidase (pdb code = 3AXH) [41] in the presence of isomaltase was downloaded from the Protein Data Bank [53]. Therefore, the amino acid sequence of the biological target was aligned with the corresponding residues of 3AXH, thanks to the Blosum62 matrix implemented in the MOE software [51]. The reliability of the alignment derived was verified by the high value of the pairwise percentage residue identity (PPRI) evaluated between the two enzymes (PPRI = 72%). Among the homology models calculated by MOE, the best scored was further optimized by full energy minimization using the AMBER94 force field [54]. The backbone conformation of the obtained final model was inspected by the Ramachandran plot, thanks to MOE software, showing the absence of outliers.

The search for the most probable binding domains within the modeled protein was performed by MOE Site Finder, suggesting the three top-ranked cavities OBS, ABS1, and ABS2. The purpose of Site Finder is to calculate possible active sites in a receptor from the 3D atomic coordinates of the receptor. The MOE site Finder falls into the category of geometric methods, taking into account the relative positions and accessibility of the receptor atoms on the basis of a classification of chemical types. The Site Finder methodology works on the Alpha Shapes represented by convex hulls [55]. The module of this software then identifies regions of tight atomic packing to filter out sites that are “too exposed” to solvent. After the protrusions that are unlikely to be good active sites are removed, the calculated preliminary sites are subsequently classified on the basis of their hydrophobic or hydrophilic profile. This coarse classification of chemical types is used to separate water sites from more likely hydrophobic sites. Afterward, the generation of alpha spheres on these preliminary sites has to be performed by eliminating those that correspond to inaccessible or are too exposed to the solvent regions of the protein [56]. Finally, all collected sites based on hydrophilic/hydrophobic properties are ranked according to their Propensity for Ligand Binding (PLB) score, which is based on the amino acid composition of the pocket [57]. Calculations supporting docking studies were performed using the DOCK tool implemented in MOE, choosing as binding sites the previously mentioned OBS, ABS1, and ABS2 [58]. Briefly, the alpha triangle placement algorithm was selected, run by superposition of ligand-atom triplets and triplets of receptor site points. The receptor site points are represented by alpha sphere centers. At each iteration, a random conformation is selected. A random triplet of ligand atoms and a random triplet of alpha-sphere centers are used to determine the pose. Calculation of the affinity G scoring function based on enthalpy allowed us to score the 50 generated poses, while the induced fit method has been used to refine the previous poses to the final ten docking poses. These were scored on the basis of the Alpha HB methodology and on the basis of H-bonding estimation. This affinity ∆G function estimates the enthalpy contribution to the free energy of binding using a linear function: ∆G = C_hb_ f_hb_ + C_ion_ f_ion_ + C_mlig_ f_mlig_ + C_hh_ f_hh_ + C_hp_ f_hp_ + C_aa_ f_aa_, where the f terms fractionally count atomic contacts of specific types and C are the coefficients that weight the term contributions to the affinity estimate (Table 6).

The induced fit approach allows the maintenance of flexible protein side chains within the selected binding sites, which will be included in the refinement stage. The derived docking poses were prioritized by the score values of the lowest energy pose of the compounds docked to the protein structure (Table 7).

### 3.7. Statistical Analysis

Data are expressed as mean ± SEM of 3–6 independent experiments. All data were analyzed using Microsoft Excel for Windows 10 and GraphPad Prism 6.0 0 (San Diego, CA, USA). The half maximal inhibitory concentration (IC_50_) was estimated by nonlinear regression. Statistical comparisons among three or more groups were performed using one-way ANOVA, followed by Tukey’s multiple comparison test. The level of significance was established at *p* < 0.05.

## 4. Conclusions

This research shows, for the first time, that magnolol efficiently inhibits α-glucosidase activity, with at least a five-fold greater potency than acarbose, exhibiting a reversible mixed-type interaction. Docking studies indicate that magnolol can interact with the SB2 site of α-glucosidase through its polar functionalities, while hydrophobic bonds can drive the enzyme inhibition. Furthermore, the data further support the role of luteolin as a non-competitive reversible inhibitor with high potency and efficacy against α-glucosidase. Therefore, now there are two possibilities to search for other effective α-glucosidase inhibitors, exploring (a) new types of substitutions, such as magnolol and luteolin scaffold decorations, and (b) new core structures that preserve the main chemical characteristics of the two parent drugs. Furthermore, other in vitro studies on mammalian α-glucosidase and in vivo studies in diabetes models are required to develop magnolol and luteolin as new potential antihyperglycemic agents.

## Figures and Tables

**Figure 1 pharmaceuticals-15-00205-f001:**
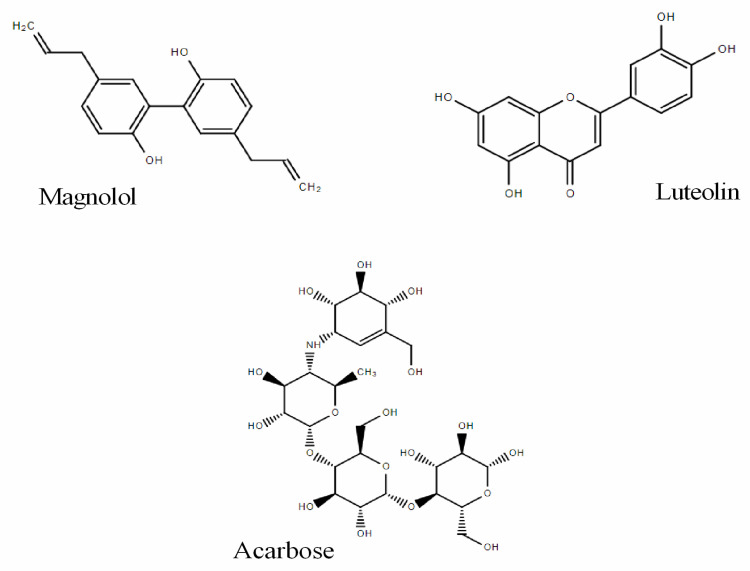
Chemical structures of magnolol and luteolin and the reference compound acarbose.

**Figure 2 pharmaceuticals-15-00205-f002:**
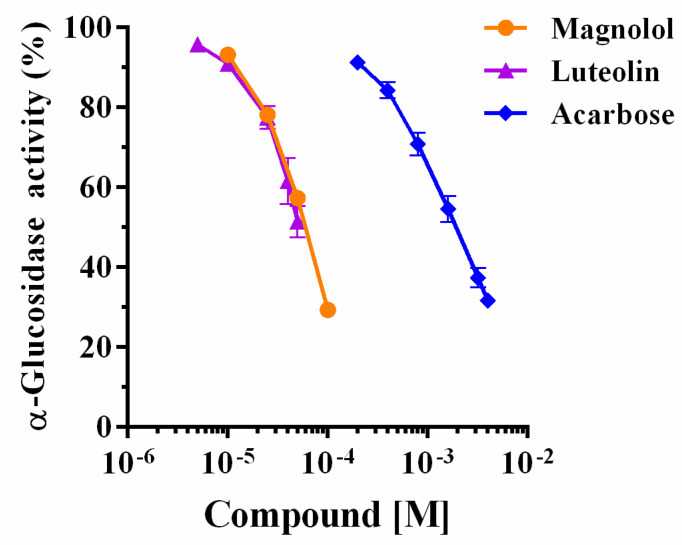
Glucosidase activity expressed as a percentage of maximal activity in the presence of magnolol, luteolin, and acarbose.

**Figure 3 pharmaceuticals-15-00205-f003:**
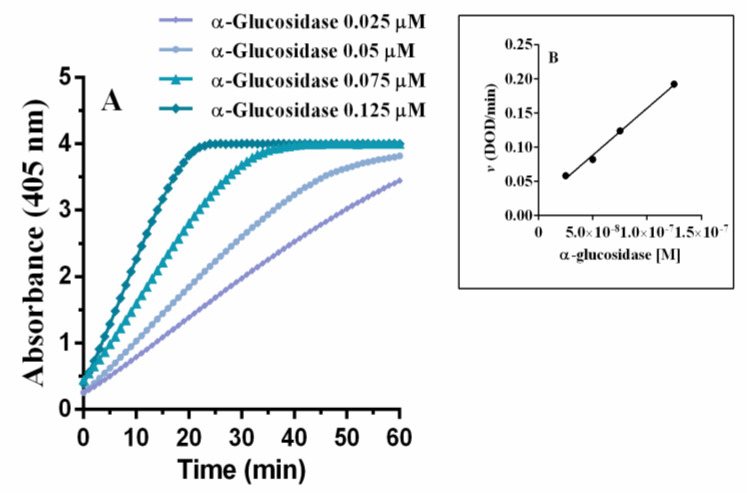
Enzyme kinetics observed with different concentrations of α-glucosidase and 2 mM pNPG (**A**). Insert: graph “*v versus* [α-glucosidase]” (**B**).

**Figure 4 pharmaceuticals-15-00205-f004:**
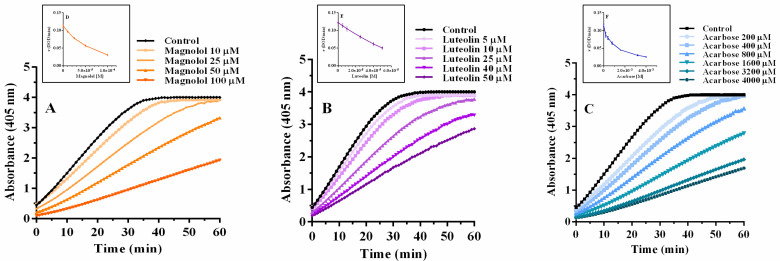
Enzyme kinetics of 0.075 µM α-glucosidase in the presence of increasing concentrations of magnolol (**A**), luteolin, (**B**) and acarbose (**C**).

**Figure 5 pharmaceuticals-15-00205-f005:**
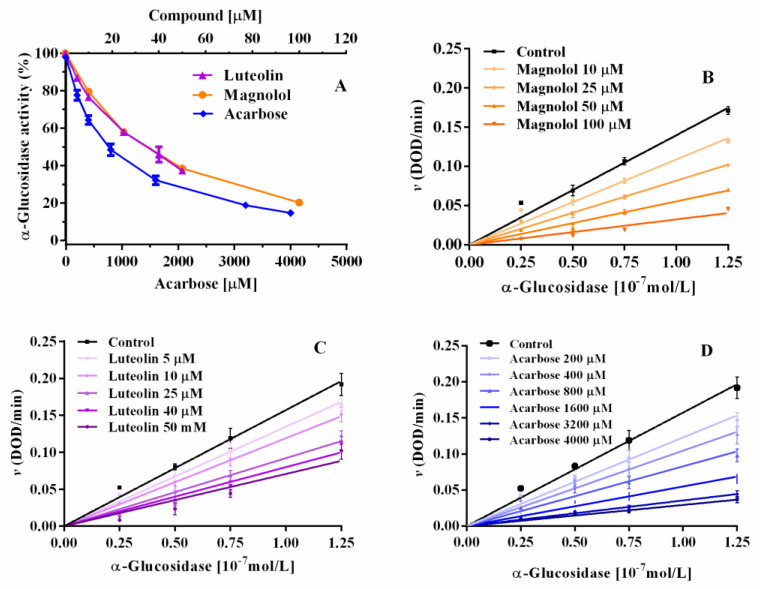
α-Glucosidase activity in the presence of magnolol, luteolin, and acarbose (linear scale) (**A**). The plots “*v versus* [α-glucosidase]” of magnolol (**B**), luteolin (**C**), and acarbose (**D**) are reported. α-Glucosidase concentration: 0.075 µM, pNPG concentration: 2 mM.

**Figure 6 pharmaceuticals-15-00205-f006:**
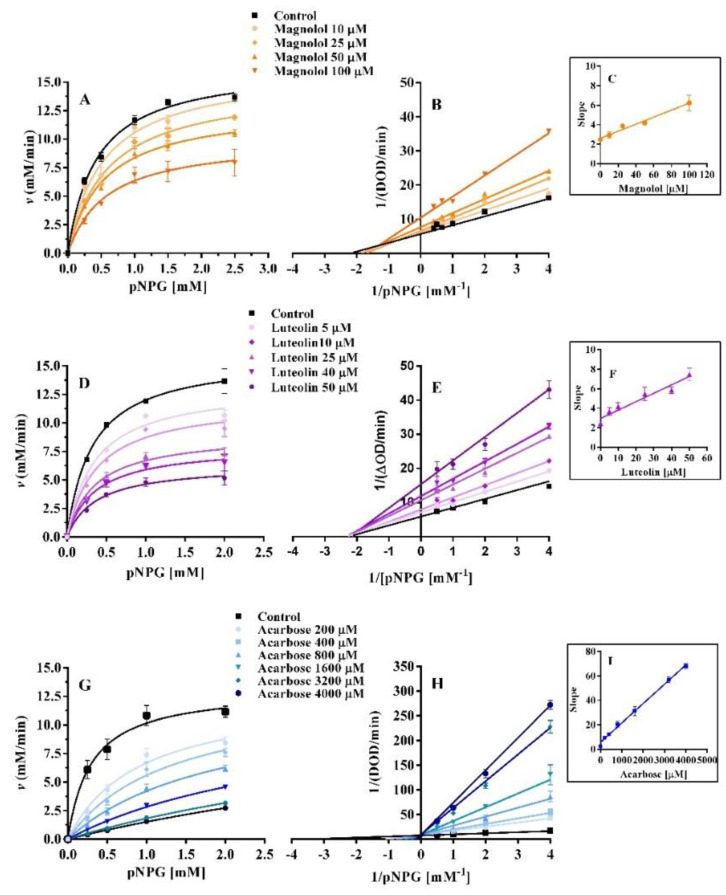
Michaelis–Menten and Lineaweaver–Burk graphs of magnolol (**A**,**B**), luteolin (**D**,**E**), and acarbose (**G**,**H**) of α-glucosidase activity with different substrate concentrations (0.25–2.5 mM). Graphs (**C**,**F**,**I**) are the secondary plots of “slope *versus* magnolol, luteolin, and acarbose concentrations”.

**Figure 7 pharmaceuticals-15-00205-f007:**
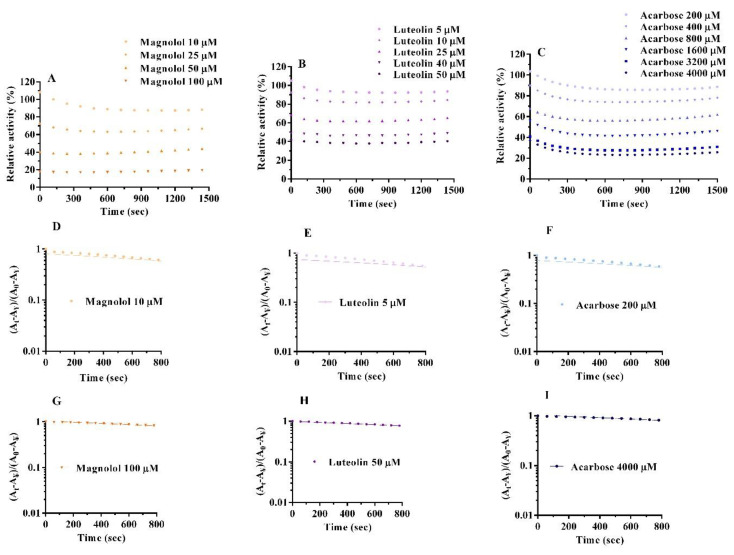
Kinetic time courses for the relative activity of 0.75 µM α-glucosidase with 2 mM pNPG in the presence of magnolol (**A**), luteolin, (**B**) and acarbose (**C**). Semilogarithmic plot analysis for magnolol (**D**,**G**), luteolin (**E**,**H**), and acarbose (**F,I**) considering the lowest and highest concentrations.

**Figure 8 pharmaceuticals-15-00205-f008:**
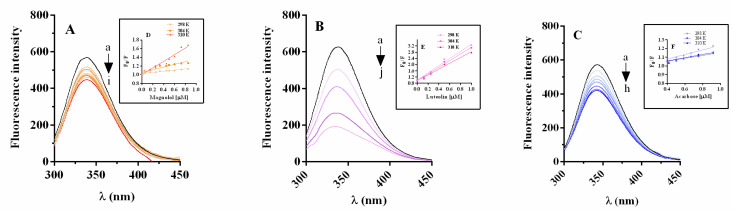
Fluorescence spectra of α-glucosidase in the presence of magnolol (**A**), luteolin (**B**), and acarbose (**C**), pH = 6.8 at 298 K. Concentrations of: α-glucosidase 0.35 µM, magnolol 0, 0.045, 0.15, 0.25, 0.35, 0.45, 0.50, 0.60, 0.725 µM, luteolin 0, 0.125, 0.25, 0.5, 1, 5, 7.5, 10 µM, and acarbose 0, 0.42, 0.503, 0.586, 0.669, 0.752, 0.918, 1.08, for curves a → i, a → j, a → h. Insets: Stern–Volmer plots for fluorescence quenching of α-glucosidase with magnolol (**D**), luteolin (**E**) and acarbose (**F**) at 298, 304 and 310 K.

**Figure 9 pharmaceuticals-15-00205-f009:**
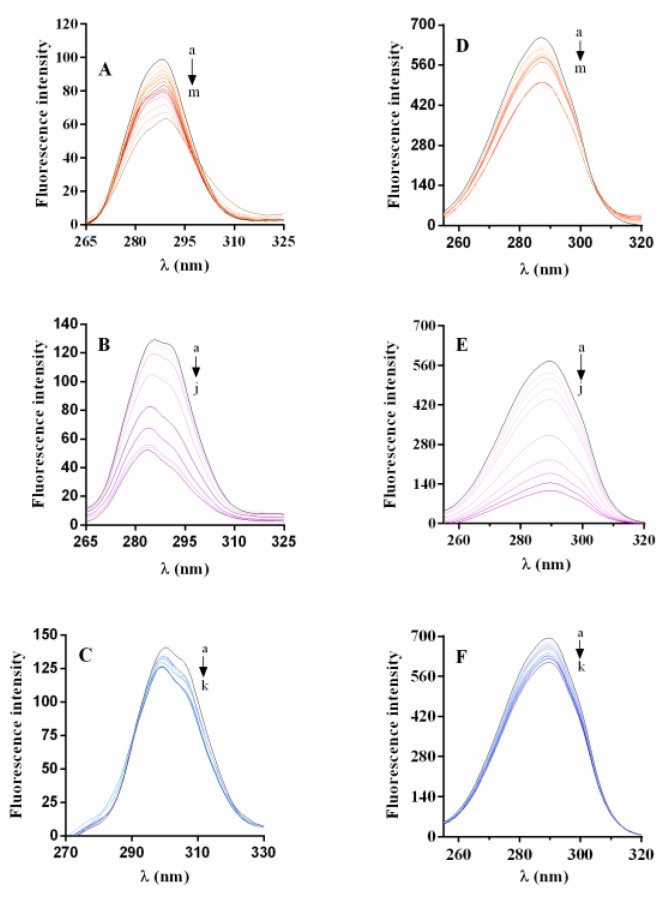
Synchronous fluorescence spectra of α-glucosidase with inhibitors (pH 6.8, T = 298 K) at ∆λ = 15 nm, magnolol (**A**), luteolin (**B**), and acarbose (**C**), and at ∆λ = 60 nm, magnolol (**D**), luteolin (**E**), and acarbose (**F**). Concentrations of: magnolol 0.0, 0.045, 0.15, 0.25, 0.35, 0.45, 0.50, 0.60, 0.725, 0.85, 0.95 µM; luteolin 0.0, 0.125, 0.25, 1, 5, 7.5, 10, 12,5, 15 µM, and acarbose 0.0, 0.42, 0.503, 0.586, 0.669, 0.752,0.918,1.08, 1.23, 6.4, 11.56, 22 µM, for curves a → m, a → j, a → k, respectively.

**Figure 10 pharmaceuticals-15-00205-f010:**
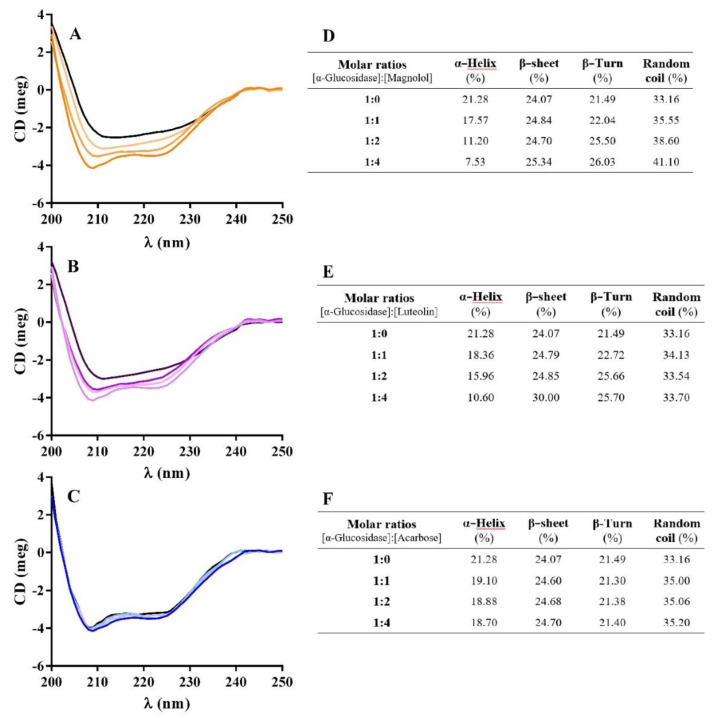
Circular dichroism spectra of α-glucosidase (1 µM) in the presence of increasing concentrations of magnolol (**A**), luteolin (**B**), and acarbose (**C**). Inserts: secondary structure contents (**D**–**F**; pH = 6.8).

**Figure 11 pharmaceuticals-15-00205-f011:**
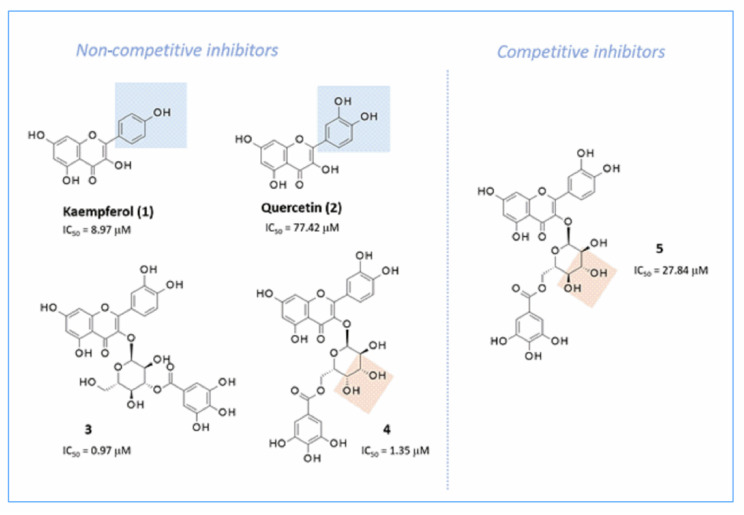
Chemical structures and inhibitory activities of chromones without sugar kaempferol (**1**) and quercetin (**2**) and of compounds **3**–**5** based on sugar (O-galloyl) as α-glucosidase inhibitors.

**Figure 12 pharmaceuticals-15-00205-f012:**
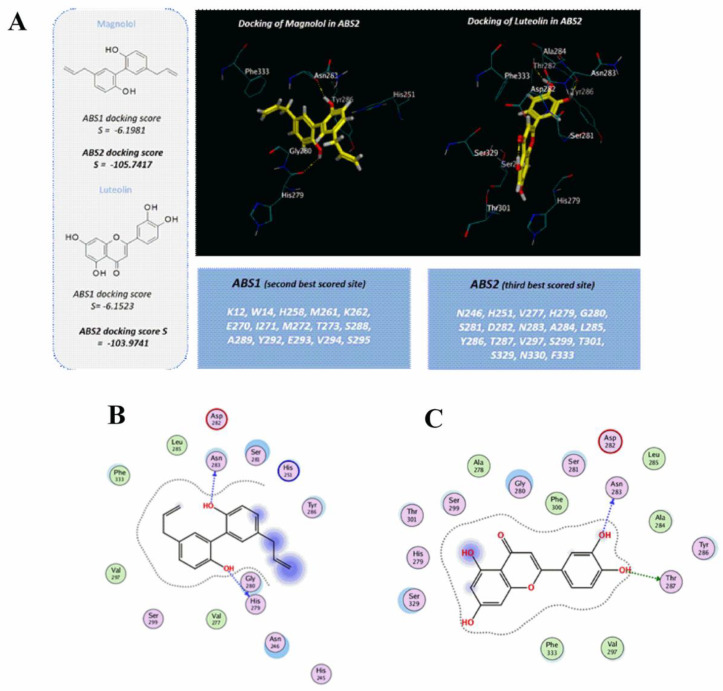
Docking positioning of the α-glucosidase inhibitors magnolol and luteolin at the ABS2 binding site (**A**). Only the most relevant residues are depicted. The amino acids included in ABS2 are listed. Ligplot (**B**) reports the most relevant contacts between α-glucosidase and magnolol in ABS2. Ligplot (**C**) reports the most relevant contacts between the α-glucosidase and luteolin in ABS2. Hydrophobic and polar residues are shown in green and pink, respectively, while the negative- and positive-charged amino acids are detailed by red and blue circles.

**Table 1 pharmaceuticals-15-00205-t001:** Inhibitory potency of magnolol, luteolin, and acarbose as inhibitors of α-glucosidase activity.

α-Glucosidase10^−6^ M	Magnolol IC_50_		Luteolin IC_50_		Acarbose IC_50_	
	10^−5^ M	−Log IC_50_	10^−5^ M	−Log IC_50_	10^−5^ M	−Log IC_50_
0.025	2.83 ± 1.07 ^a^	4.55 ± 0.03	1.40 ± 1.05 ^a^	4.85 ± 0.02	80.54 ± 1.02 ^b^	3.09 ± 0.01
0.05	2.85 ± 1.10 ^a^	4.55 ± 0.04	2.16 ± 1.17 ^a^	4.66 ± 0.07	86.80 ± 1.05 ^b^	3.06 ± 0.02
0.075	3.26 ± 1.05 ^a^	4.49 ± 0.02	3.23 ± 1.17 ^a^	4.49 ± 0.07	81.54 ± 1.12 ^b^	3.09 ± 0.05
0.125	3.55 ± 1.02 ^a^	4.26 ± 0.01	5.94 ± 1.23 ^a^	4.23 ± 0.09	80.70 ± 1.10 ^b^	3.09 ± 0.04

IC_50_: half maximal inhibitory concentration of enzyme activity; M: molarity. Data are obtained using the non-linear regression of the normalized response of 3–7 experiments. Values not sharing a common superscript letter differ significantly at *p* < 0.0001 tested by one-way ANOVA, followed by Tukey’s multiple comparison test.

**Table 2 pharmaceuticals-15-00205-t002:** The kinetic parameters of magnolol, luteolin, and acarbose on α-glucosidase activity and relative inhibitory potency.

Compound	Concentration (µM)	K_m_ (mM)	K_cat_ (sec^−1^)	K_cat_/K_m_ (sec^−1^ mM^−1^)	K_i_ (µM)	K_i’_ (µM)	Inhibitor Potency
Magnolol	0	0.48 ± 0.054	3.89 ± 0.14	8.18 ± 2.62	78.3	132.4	4.5
	10	0.53 ± 0.08	3.59 ± 0.17	6.78 ± 2.17			
	25	0.59 ± 0.08	3.29 ± 0.15	5.54 ± 1.82			
	50	0.54 ± 0.08	2.86 ± 0.13	5.52 ± 1.55			
	100	0.58±0.24	2.22 ± 0.29	3.87 ± 1.26			
Luteolin	0	0.32 ± 0.04	3.52 ± 0.14	11.01 ± 0.43	34.2	35.4	10.4
	5	0.35 ± 0.05	2.95 ± 0.12	8.43 ± 0.32			
	10	0.37 ± 0.05	2.65 ± 0.13	7.17 ± 0.34			
	25	0.36 ± 0.07	2.02 ± 0.12	5.60 ± 0.34			
	40	0.36 ± 0.07	1.78 ± 0.12	4.94 ± 0.33			
	50	0.37 ± 0.05	1.41 ± 0.11	3.81 ± 0.31			
Acarbose	0	0.30 ± 0.08	2.92 ± 0.23	9.75 ± 2.83	356.3	_	1
	200	0.91 ± 0.23	2.83 ± 0.33	3.11 ± 1.42			
	400	1.31 ± 0.34	2.87 ± 0.39	2.19 ± 1.14			
	800	1.96 ± 0.52	2.77 ± 0.44	1.41 ± 0.84			
	1600	3.24 ± 0.64	2.66 ± 0.36	0.82 ± 0.56			
	3200	5.91 ± 1.81	2.79 ± 0.67	0.47 ± 0.37			
	4000	8.30 ± 3.32	3.12 ± 1.04	0.38 ± 0.36			

α-Glucosidase concentration: 0.075 µM; pNPG concentration: 2 mM; K_m_: Michaelis–Menten constant; K_cat_: enzyme cycle per unit of time; K_i_: binding constant of the inhibitor to the free enzyme; K_i’_: binding constant of the inhibitor to the substrate-enzyme complex; K_cat_/K_m_: catalytic efficiency of α-glucosidase. Constant values were obtained using the data of Figure 6.

**Table 3 pharmaceuticals-15-00205-t003:** Inactivation rate constants (K) and transition free energy change (ᴧᴧG°) of magnolol, luteolin, and acarbose on α-glucosidase.

Compound	Concentration (µM)	K (10^−4^ s^−1^)	ᴧᴧG° (kJ mol^−1^ s^−1^)
Magnolol	10	2.78 ± 0.05	21.10
	25	2.81 ± 0.03	21.08
	50	2.81 ± 0.01	21.07
	100	2.85 ± 0.08	21.04
Luteolin	5	2.67 ± 0.08	21.21
	10	2.78 ± 0.08	21.10
	25	2.81 ± 0.03	21.07
	40	2.82 ± 0.04	21.06
	50	2.84 ± 0.01	21.05
Acarbose	200	2.76 ± 0.07	21.12
	400	2.81 ± 0.07	21.07
	800	2.83 ± 0.05	21.06
	1600	2.84 ± 0.03	21.05
	3200	2.85 ± 0.01	21.04
	4000	2.85 ± 0.09	21.03

ᴧᴧG° = −RT ln K, where K is the time constant for the monophasic inactivation reaction.

**Table 4 pharmaceuticals-15-00205-t004:** The quenching constant (K_sv_) and the bimolecular quenching constant (K_q_) of magnolol, luteolin, and acarbose with α-glucosidase at different temperatures.

Compound	T (K)	K_sv_ (10^5^ M^−1^)	K_q_ (10^13^ M^−1^ s^−1^)
Magnolol	298	3.89 ± 0.01	3.90 ± 0.007
	304	4.22 ± 0.01	4.22 ± 0.011
	310	5.30 ± 0.02	5.30 ± 0.022
Luteolin	298	23.22 ± 0.16	23.22 ± 0.159
	304	21.73 ± 0.14	21.73 ± 0.141
	310	18.68 ± 0.12	18.68 ± 0.120
Acarbose	298	3.75 ± 0.04	3.75 ± 0.041
	304	1.89 ± 0.01	1.89 ± 0.014
	310	1.80 ± 0.02	1.80 ± 0.020

T: temperature; K_sv_: Stern–Volmer constant; K_q_: bimolecular quenching constant.

**Table 5 pharmaceuticals-15-00205-t005:** Thermodynamic and binding parameters of magnolol, luteolin, and acarbose of the α-glucosidase interaction obtained at different temperatures.

System	T (K)	K_a_ (10^5^ M^−1^)	n	ΔG° (kJ mol^−1^)	∆S° (J mol^−1^K^−1)^	ΔH° (kJ mol^−1^)
Magnolol-α-glucosidase	298	2.97 ± 0.02	0.39	−2.92	121.83	39.23
	304	4.54 ± 0.02	0.42	−2.19		
	310	5.48 ± 0.02	0.65	−1.46		
Luteolin-α-glucosidase	298	24.86 ± 0.05	1.10	−2.29	−22.95	−9.13
	304	23.99 ± 0.07	1.28	−2.15		
	310	21.54 ± 0.10	1.34	−2.01		
Acarbose- α-glucosidase	298	3.91 ± 0.06	1.77	2.60	−161.47	−45.51
	304	1.94 ± 0.04	1.38	3.57		
	310	1.93 ± 0.06	1.36	4.54		

T: temperature in Kelvin scale; K_a_: binding constant; n: number of binding sites. The enthalpy change (ΔH°), entropy change (ΔS°), and free energy change (ΔG°) were determined using the Van ’t Hoff equation.

**Table 6 pharmaceuticals-15-00205-t006:** Description of the calculated terms taken into account to estimate the complex affinity ∆G value.

Subscript	Description
hb	Interactions between hydrogen bond donor–acceptor pairs. An optimistic view is taken; for example, two hydroxyl groups are assumed to interact in the most favorable way.
ion	Ionic interactions. A Coulomb-like term is used to evaluate the interactions between charged groups. This can contribute to or detract from the binding affinity.
mlig	Metal ligation. Interactions between nitrogens/sulfurs and transition metals are assumed to be metal-ligation interactions.
hh	Hydrophobic interactions, for example, between alkane carbons. These interactions are generally favorable.
hp	Interactions between hydrophobic and polar atoms. These interactions are generally unfavorable.
aa	An interaction between two atoms. This interaction is weak and generally favorable.

**Table 7 pharmaceuticals-15-00205-t007:** Description of the main scoring functions employed for MOE molecular docking calculations.

S	The Final Score, Which Is the Score of the Last Stage of Refinement.
E_conf	The energy of the conformer. If there is a refinement stage, this is the energy calculated at the end of the refinement
E_place	Score from the placement stage
E_score1 E_score2	Score from rescoring stages 1 and 2
E_refine	Score from the refinement stage, calculated to be the sum of the van der Waals electrostatics and solvation energies, under the Generalized Born solvation model (GB/VI)

## Data Availability

Data is contained within the article and Appendix A.

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
