# Peer review of "Magnolol and Luteolin Inhibition of α-Glucosidase Activity: Kinetics and Type of Interaction Detected by In Vitro and In Silico Studies"

_pharmaceuticals, 2022, doi:10.3390/ph15020205_

Round 1

Reviewer 1 Report

1. Separate group has reported the same topic on same enzyme doi: 10.1080/14756366.2017.1368503 in year 2017. What is the novelty of present work? 2. Molecular dynamic simulation study can provide useful information in enzyme inhibitor complex formation as authors are predicting interaction of inhibitory molecules with α-glucosidase. 3. Significance and other relevant statistical parameters should be applied to figures and tables throughout the manuscript and authors must report the method that they used for calculation of these parameters 4.  Binding constant values are different from Ksv values. Why? 5. Conformational changes in proteins on inhibitor interaction were explored but not discussed properly in paper 6. Information and data are not properly presented and discussed in paper, lot of experimental data is under presented and discussion is confusing and authors are not focused in their studies  7. Molecular docking is prediction, thermodynamic data can be obtained by using ITC methods more accurately 8. Authors used DMSO, did they noticed any enzyme alterations with concentrations used in experiments during the study 9. Authors found any correlation between synchronous fluorescence data and CD data upon the interaction of inhibitors with enzyme 10. Near UV-CD data can indicate topological changes in enzymes upon the complexation of inhibitors 11. Molecular complexation has resulted in conformational alterations as indicated in CD spectra of enzyme, this may be the actual cause of inhibitory effect, do these changes has some effect on enzyme biological functions and other consequences. 12. Language and style can be improved throughout the manuscript; authors need to work on it.

Author Response

We thank the Editor and Reviewers for their valuable suggestions in order to improve our manuscript. All comments and questions have been carefully considered for the revision.

Response to Reviewer 1

Thanks for your valuable review that helped us to revise the manuscript. All suggestions have been taken into account, and all changes are reported with the revision tracking in Word.

Point 1. Separate group has reported the same topic on same enzyme doi: 10.1080/14756366.2017.1368503 in year 2017. What is the novelty of present work?

Response 1. The paper by Proença et al. (2017) presented an extensive investigation about α-glucosidase inhibition by a panel of several flavonoids, considering the structure-activity relationships. They considered, among several evaluated flavonoids, also luteolin (named D7), but only in vitro inhibition of α-glucosidase was determined, leaving a further possibility of deepening the study of the compound considered. Most importantly, this article did not consider magnolol, which was our main topic in the current investigation. Furthermore, no research has been reported in the literature on magnolol on in vitro α-glucosidase inhibition.

The article by Proença et al. (2017) had already been quoted in our manuscript and now further information on their work has been provided, also to motivate our research. For both magnolol and luteolin, we provide, in addition to the α-glucosidase kinetic parameters, thermodynamic and binding parameters, Förster resonance energy transfer, synchronous fluorescence spectra, circular dichroism spectra, and computational evaluation including analysis of the binding site and molecular coupling. Also, with respect to the paper by Yan et al. (2014) (ref. 17), our study improves knowledge about luteolin, providing new parameters and comparison with magnolol.

In Proença et al. (2017), homology modeling studies of Saccharomyces cerevisiae α-glucosidase have been performed based on the apoform of the enzyme (PDB code = 3AJ7), lacking substrates. In fact, the authors modelled the substrate α-maltose within a putative binding pocket. In current research, we refer to a different template (PDB code = 3AXH) related to 1,6-glucosidase, in the presence of isomaltose. This choice allowed us to apply a ligand-based homology (LBH) modeling strategy, in order to take into account the structure of the co-crystallized ligand isomaltose during the protein homology modeling calculation. This led to the development of a more accurate and reliable enzyme model, whose isomaltose-binding site was carefully managed. The final model represented a robust starting point for the development of the molecular docking calculations, which were derived and compared in section 2.7. with the experimental data reported in the literature [ref. 19].

Point 2 Molecular dynamic simulation study can provide useful information in enzyme inhibitor complex formation as authors are predicting interaction of inhibitory molecules with α-glucosidase.

Response 2. Molecular coupling is a good investigational approach to obtain a perspective of α-glucosidase system at the molecular level generated by magnolol and luteolin, based on preliminary inhibitory activities of sugar-free chromones, such as kaempferol and quercetin. Surely, some higher-complexity techniques such as molecular dynamics (MD) could be applied to the system, but the biological situation that has to be represented would be so different with respect to the simulation conditions, suggesting that MD application would difficulty increase the validity of the hypothesis. Therefore, even if MD studies performed on different enzyme-docked ligand complexes were desirable, this would be very time consuming and difficult for us to manage in a reasonable time. Furthermore, considering the consistent number of experimental approaches applied in the current investigation, MD is thought to be outside the scope of this work. We thank the Reviewer for the suggestion that will be taken into account in future research.

Point 3 Significance and other relevant statistical parameters should be applied to figures and tables throughout the manuscript and authors must report the method that they used for calculation of these parameters

Response 3 The statistical calculation has been reported when appropriate. For details, please refer to Table 1 (IC50 of the compounds detected). Furthermore, statistical methods have been explained in section 3.7. Statistical analysis.

Point 4.  Binding constant values are different from Ksv values. Why?

Response 4. Fluorescence quenching refers to any process that decreases the fluorescence intensity of a sample. A variety of molecular interactions can result in quenching. These include excited-state reactions, ground-state complex formation, collisional quenching, etc. The Stern-Volmer constant (Ksv) determines the kinetics of a photophysical intermolecular deactivation process and therefore can be different from the value of binding constant (Ka). In the context of the present investigation, rather than as an absolute number, Ksv should be considered as a relative comparison among different compounds, namely magnolol and luteolin versus acarbose, the latter taken as a reference. In particular, it is important to study how Ksv changes in relation to increasing temperature to evaluate the type of interaction between the enzyme (fluorescent) and the inhibitor (quencher) used at different concentrations, as reported in Section 2.5.1. Anyway, Ksv and Ka constants are derived from different formulae which were added in Section 4.4.1. However, the expression of the Ka values was changed from 106 to 105, evidencing the similarity between Ksv and Ka, which cannot be the same because they are obtained from different equations and their meaning is also dissimilar.

Point 5. Conformational changes in proteins on inhibitor interaction were explored but not properly discussed in the paper.

Response 5. A more detailed discussion is provided in Section 2.6 of the manuscript.

Point 6. Information and data are not properly presented and discussed in the paper, a lot of experimental data is not presented, and the discussion is confusing and authors are not focused in their studies.  

Response 6 The result and discussion Section has been improved trying to be clearer.

Point 7. Molecular docking is a prediction, and thermodynamic data can be obtained by using ITC methods more accurately.

Response 7 We agree with the Reviewer on the theoretical information coming from docking studies. The reliability of these calculations has been assessed herein by comparison of the binding site analyses and of the docking poses obtained for compounds shown in Figure 11 with respect to the information published in ref. 19. Isothermal titration calorimetry (ITC) would give specific, experimentally-linked results of the phenomena, but the technique is not available to us. However, the pool of methods used in our investigation may provide information that allows an overall evaluation of the ability of the inhibitor to reduce enzyme activity.

Point 8. Authors used DMSO, did they noticed any enzyme alterations with concentrations used in experiments during the study?

Response 8. DMSO was used at a very low concentration (<1% v/v, Section 3.2.), and no appreciable changes in enzyme activity were observed up to this concentration.

Point 9. Authors found any correlation between synchronous fluorescence data and CD data upon the interaction of inhibitors with enzyme

Response 9. Indeed, a relation was detected between synchronous fluorescence and CD data. Synchronous fluorescence data indicated a change in the microenvironment of tyrosine, suggesting that magnolol and luteolin affect the enzyme structure, leading to exposure to solvent and subsequent displacement of the tyrosine residues to more hydrophilic residues. This effect can be linked to the change in the structure of α-helix, found with CD, since tyrosine, which has a large side group, is able to destabilize α-helices. This has been mentioned in Sections 2.6.1 and 2.6.2 of the manuscript.

Point 10. Near-UV-CD data can indicate topological changes in enzymes upon the complexation of inhibitors.

Response 10. Indeed, the CD data may suggest changes in ligand-enzyme topology, also in a concentration-dependent manner. Furthermore, according to the following suggestion No.11, further discussion has been provided in the manuscript.

Point 11. Molecular complexation has resulted in conformational alterations as indicated in CD spectra of the enzyme; this may be the actual cause of inhibitory effect, but do these changes have some effect on enzyme biological functions and other consequences.

Response 11. As suggested by CD analyses performed on α-glucosidase inhibition by tannins (Ma et al., 2015, doi:10.1039/C5RA19014B), binding of the enzyme ligand leads to loss of the secondary α-helix structure of the protein, which could cause reduction of enzyme biological activity. A mention of this fact has been added in the manuscript.

Point 12. Language and style can be improved throughout the manuscript; authors need to work on it.

Response 12. A careful check of the language and style has been performed.

Reviewer 2 Report

The work of Djeujo et al. concerns the in vitro and in silico study of the enzymatic activity of two natural compounds Magnolol and luteolin and their activity on the α-glucosidase.

The manuscript is well structured and complete in its parts highlighting extensive in vitro assay work.

In addition, the research topic is well suited for the special issue “Bioactive Compounds from Plants and Foods with Pharmaceutical Interest 2022”, and, for these reasons, I recommend its publication.

Minor observations, which do not influence the overall quality, but which could further improve the reading, concern the homology modeling section.

The authors in the absence of the three-dimensional structure of the α-glucosidase protein decided to build it using the homology modeling method. It is unclear what software was used in this step (add proper citation) and especially unclear how many templates were used to predict the three-dimensional structure.

However, in section 4.6 only the 3AXH PDB is reported suggesting that the structure was forced to be built on this one structure (the % homology sequence value is missing).

 The authors should add the missing information and better describe this part.

Author Response

Manuscript pharmaceuticals-1553189 

We thank the Editor and Reviewers for their valuable suggestions in order to improve our manuscript. All comments and questions have been read with attention and considered for revision.

Response to Reviewer 2

Thanks for your appreciated review that helped us to revise the manuscript. The suggestions have been considered, and all changes are reported with the revision tracking in Word.

“The work of Djeujo et al. concerns the in vitro and in silico study of the enzymatic activity of two natural compounds Magnolol and luteolin and their activity on the α-glucosidase. The manuscript is well structured and complete in its parts highlighting extensive in vitro assay work. In addition, the research topic is well suited for the special issue “Bioactive Compounds from Plants and Foods with Pharmaceutical Interest 2022”, and, for these reasons, I recommend its publication.

Minor observations, which do not influence the overall quality, but which could further improve the reading, concern the homology modeling section.”

Point 1. The authors in the absence of the three-dimensional structure of the α-glucosidase protein decided to build it using the homology modeling method. It is unclear what software was used in this step (add proper citation) and especially unclear how many templates were used to predict the three-dimensional structure.

Response 1. The required information has been clarified throughout the manuscript. In detail, protein modeling was performed on the basis of ligand-based homology (LBH) modeling approaches. We only refer to the experimental data of 1,6-glucosidase (pdb code= 3AXH) [ref. 42; doi:10.1016/j.jbiosc.2011.08.016.] in the presence of isomaltose, as a template for the in-house α-glucosidase model.

The theoretical model of the biological target has been built using MOE software [ref. 51; MOE: Chemical Computing Group Inc. Montreal. H3A 2R7 Canada. http://www.chemcomp.com.] by alignment of the α-glucosidase FASTA sequence (P38158), as downloaded from the SWISSPROT database [ref. 52, doi:10.1093/nar/28.1.45], with respect to the X-ray coordinates of the previously quoted template (pdb code= 3AXH). In particular, the BLOSUM62 alignment matrix has been exploited. Homology modeling studies have been better detailed in the material and methods section (3.6.), clarifying the exploited software and template. The required information is detailed in the section.

Point 2 However, in Section 4.6 only the 3AXH PDB is reported, suggesting that the structure was forced to be built on this one structure (the % homology sequence value is missing).

Response 2. We apologize for the inconvenience. The section was more detailed. In fact, the modelled protein was built using MOE software using the X-ray coordinates of 1,6-glucosidase as a single template (pdb code= 3AXH). Thanks for the suggestion.

Point 3. The authors should add the missing information and better describe this part.

Response 3. We thank the Reviewer for the suggestion. The theoretical model of the biological target has been built using MOE software [ref. 51; MOE: Chemical Computing Group Inc. Montreal. H3A 2R7 Canada. http://www.chemcomp.com] using the X-ray coordinates of 1,6-glucosidase as a single template (pdb code= 3AXH). Alignment of the two protein primary sequences has been done using the BLOSUM62 matrix implemented in MOE.

Results and Discussion section (2.7.) has been improved accordingly and detailed, also using the supporting information S3 A, B and C. In particular, the alignment reliability of the obtained alignment was verified by the high value of the pairwise percentage of residue identity evaluated between the two enzymes (PPRI=72%, Figure S3 A). Therefore, a consistent number of residues was conserved between the two proteins, as reported by superposition of the modeled protein with respect to the template (Figure S3 B), which has an adequate root mean square deviation value (RMSD = 0.229 Å). The backbone conformation of the final yeast α-glucosidase final model was also inspected by the Ramachandran plot, showing the absence of outliers (Figure S3 C). Results and Discussion section (2.7.) has been improved accordingly and detailed, also using the supporting information S3 A, B and C.

Reviewer 3 Report

The study of Djeujo et al. represents an interdisciplinary work aimed to characterize the potency of magnalol and luteolin, natural plant-derived compounds, as α-glycosidase inhibitors. The authors apply in vitro (inhibitory assays, kinetic analysis, non-radiation energy transfer, fluorescence, circular dichrosim) and in silico (homology modeling and molecular docking) approaches to characterize the inhibitory effect of these molecules and compare it with the well-established α-glycosidase inhibitor acarbose. In particular, the types of inhibition of these three compounds as well as their kinetic mechanisms were rigorously analyzed and discussed, the type of interactions (in terms of thermodynamics and physical forces involved) were established, the impact of the inhibitor assciation with the enzyme on the enzyme structural properties was revealed and putative binding sites (involved amino acid residues) were proposed by molecular modeling. The methods utilized in the study are properly applied and extensively described, the manuscript is clearly and logically written, the conclusions are relevant. The data obtained in this work could serve as a basis for the further rational design of new highly specific inhibitors of α-glycosidase, which could contribute to the novel approaches to treat diabetes mellitus. The manuscipt could be published in Pharmaceuticals after two very small points are addressed by the authors. 

MINOR POINTS:

– the thermodynamic spontaneity should be defined/explained more clealy for the general audience (for example in 2.3.3.2 it is not clear if it is related to the sign of the enthalpy or the binding free energy);

– “α-glycosidase” and “α-Glycosidase” are inconsistenly used through the manuscript.

Author Response

We thank the Editor and Reviewers for their valuable suggestions in order to improve our manuscript. All comments and questions have been read with attention and considered for revision.

Response to Reviewer 3

Thanks for your appreciated review that helped us to revise the manuscript. The suggestions have been taken into account, and all changes are reported with the revision tracking in Word.

The study of Djeujo et al. represents an interdisciplinary work aimed at characterizing the potency of magnolol and luteolin, natural plant-derived compounds, as α-glycosidase inhibitors. The authors apply in vitro (inhibitory assays, kinetic analysis, non-radiation energy transfer, fluorescence, circular dichroism) and in silico (homology modeling and molecular docking) approaches to characterize the inhibitory effect of these molecules and compare it with the well-established α-glycosidase inhibitor acarbose. In particular, the types of inhibition of these three compounds as well as their kinetic mechanisms were rigorously analyzed and discussed, the type of interactions (in terms of thermodynamics and physical forces involved) were established, the impact of the inhibitor association with the enzyme on the enzyme structural properties was revealed and putative binding sites (involved amino acid residues) were proposed by molecular modeling. The methods utilized in the study are properly applied and extensively described, the manuscript is clearly and logically written, the conclusions are relevant. The data obtained in this work could serve as a basis for the further rational design of new highly specific inhibitors of α-glycosidase, which could contribute to the novel approaches to treat diabetes mellitus. The manuscript could be published in Pharmaceuticals after two very small points are addressed by the authors.

Point 1. – the thermodynamic spontaneity should be defined/explained more clearly for the general audience (for example in 2.3.3.2 it is not clear if it is related to the sign of the enthalpy or the binding free energy);

Response 1. An explanation of the thermodynamic spontaneity has been provided in the suggested section, now numbered 2.5 Thermodynamic parameters and nature of binding forces.

Point 2 α-glycosidase” and “α-Glycosidase” are inconsistently used through the manuscript.

Response 2. We apologize for the inconvenience. Corrections have been made to the manuscript, maintaining the term “α-glucosidase” where appropriate. We maintain the appearance “α-Glucosidase” when the term is at the beginning of a sentence or in a title, according to the editorial indications.